# Analyses of hyphal diversity in Trichosporonales yeasts based on fluorescent microscopic observations

Keita Aoki,[1] Moriya Ohkuma,[2] Takashi Sugita,[3] Yuuki Kobayashi,[1] Naoto Tanaka,[4] Masako Takashima[1]

**ABSTRACT** In dimorphic yeasts, hyphal growth is primarily associated with infection and mycosis progression, with *Trichosporon asahii* causing deep-seated mycosis and summer-type hypersensitivity pneumonitis. Magnesium accelerates hyphal growth in *T. asahii*, leading to multi-septation, vacuolar expansion, and decreased lipid droplet size. However, the commonality of these phenotypes has not been studied in Trichosporonales yeasts. Therefore, to explore whether similar magnesium-induced phenotypes occur across Trichosporonales yeasts, we examined hyphal growth, multi-septation, vacuolar extension, and lipid droplet size and number in 30 species. Cell length increased with magnesium treatment in 13 yeasts: 5 *Trichosporon* (*T. asahii*, *Trichosporon aquatile*, *Trichosporon asteroides*, *Trichosporon coremiiforme*, and *Trichosporon ovoides*), three *Apiotrichum* (*Apiotrichum brassicae*, *Apiotrichum montevideense*, and *Apiotrichum veenhuisii*), three *Cutaneotrichosporon* (*Cutaneotrichosporon cavernicola*, *Cutaneotrichosporon cutaneum*, and *Cutaneotrichosporon dermatis*), *Pascua guehoae*, and *Takashimella koratensis*. *C. dermatis* also underwent pseudo-hyphal growth. Multi-septation increased in seven dimorphic yeasts, including five *Trichosporon* spp., *Trichosporon faecale*, and *C. dermatis*. The vacuolar area was significantly extended in *T. asahii*, *T. aquatile*, *T. ovoides*, and *C. dermatis*. Lipid droplet size increased only in *Trichosporon inkin*; however, it decreased in *T. asahii*, *T. coremiiforme*, and *T. faecale*. Additionally, lipid droplet number was preferentially altered in *Apiotrichum* and *Cutaneotrichosporon*. These results suggested that magnesium-induced multi-septation and vacuolar area expansion phenotypically distinguish *Trichosporon* hyphae from *Apiotrichum* and *Cutaneotrichosporon* hyphae and distinguish *C. dermatis* pseudo-hyphae from *Cutaneotrichosporon* spp. Moreover, differences in lipid droplet size can discriminate species within *Trichosporon*. Our results suggest that phenotypic alteration via magnesium treatment can contribute to the characterization of Trichosporonales yeasts. These findings provide insights into fungal pathogenesis and may support new treatment strategies.

**IMPORTANCE** Magnesium sulfate considerably affects hyphal growth and cellular organization in *Trichosporon asahii*. To examine the commonality of this phenotype in Trichosporonales, we treated 30 Trichosporonales yeasts with magnesium sulfate and observed genus-level phenotypic alterations. Using cell length measurement, lipid droplet staining, septum staining, and vacuole staining, considerable hyphal diversity was observed among Trichosporonales. Notably, differences in the multi-septation phenotype and vacuolar size distinguished *Trichosporon* hyphae from *Apiotrichum* and *Cutaneotrichosporon* hyphae and distinguished *Cutaneotrichosporon dermatis* from other *Cutaneotrichosporon* spp. Moreover, differences in lipid droplet phenotype divided *Trichosporon* hyphae into two groups. Our study revealed the relationship between hyphal morphology and phylogenetic systematics in Trichosporonales.

**KEYWORDS** Trichosporonales, hyphal diversity, fungal dimorphism

**Peer Reviewers** Wencheng Zhu, Chinese Academy of Sciences, Shanghai, China; Marwan Yaseen Al-Maqtoofi, University of Basrah, Basrah, Basrah, Iraq

Address correspondence to Keita Aoki, ka207755@nodai.ac.jp, Keita Aoki, ka207755@nodai.ac.jp, or Masako Takashima, mt207623@nodai.ac.jp.

The authors declare no conflict of interest.

See the funding table on p. 19.

Yeasts can exist in both unicellular and multi-cellular states, adapting their morphology to extracellular environments (1). This morphological flexibility, or dimorphism, includes both unicellular yeast forms and multi-cellular hyphae, which contribute to invasion of cellular organisms in ascomycetes and basidiomycetes (2, 3). Trichosporonales is an order in basidiomycetes mainly comprising the genera *Trichosporon*, *Apiotrichum*, and *Cutaneotrichosporon*, which include dimorphic yeasts. The genome of 27 species in Trichosporonales has been sequenced. Phylogenetic relationships have been inferred based on the D1/D2 domain of the LSU rRNA gene and region including internal transcribed spacer sequence and the 5.8S rRNA gene (4). Historically, each genus in Trichosporonales was classified based on serotype, ubiquinone type, and arthroconidium (4, 5). Ubiquinone type is well characterized for classifying genera. *Trichosporon* and *Apiotrichum* are dominated by coenzyme Q9, whereas *Cutaneotrichosporon* is dominated by coenzyme Q10. Arthroconidium is abundant in *Trichosporon* and *Apiotrichum* and some *Cutaneotrichosporon* spp. that were formerly identified as *Trichosporon* (4, 5). Arthroconidium does not appear in *Cutaneotrichosporon* spp. that were formerly identified as *Cryptococcus* (4, 5). Therefore, arthroconidium presence can distinguish *Cutaneotrichosporon* into two groups. Serotyping classifies Trichosporonales yeasts into three groups, *Cutaneotrichosporon* showing arthroconidium as serotype I, *Trichosporon* as serotype II, and *Apiotrichum* genus as serotype III (4). Sugita et al. (6) identified *T. asahii* as the primary antigen in summer-type hypersensitivity pneumonitis (SHP), with nine other Trichosporonales yeasts: *Trichosporon aquatile*, *Trichosporon coremiiforme*, *Trichosporon faecale*, *Trichosporon japonicum*, *Trichosporon ovoides*, *Apiotrichum domesticum*, *Apiotrichum montevideense*, *Cutaneotrichosporon dermatis*, and *C. terricola*, also implicated in SHP. These Trichosporonales yeasts were classified into three groups via serotyping: serotype I (*C. dermatis* and *C. terricola*), serotype II (*T. asahii*, *T. aquatile*, *T. coremiiforme*, *T. faecale*, *T. japonicus*, and *T. ovoides*), and serotype III (*A. domesticum* and *A. montevideense*) (6). Therefore, these phenotypes characterize Trichosporonales yeasts at the genus level. However, microscopic observations of the morphological phenotypes of Trichosporonales yeasts remain limited.

*Trichosporon asahii* is a basidiomycetous yeast that can alternate among yeast, hyphal, and arthroconidia forms and occasionally causes trichosporonosis, a deep-seated mycosis in immunocompromised patients (7–11). Among the distinct morphological forms of *T. asahii*, hyphae are associated with infection (12), whereas arthroconidia contribute to biofilm formation by promoting cell adhesion (13). In *T. asahii* cultures, yeasts, hyphae, and arthroconidia coexist. Cell length shortens in *T. asahii* in Sabouraud broth and notably extends when supplemented with magnesium, indicating magnesium-dependent hyphal formation (MagHF) in *T. asahii* (14). The MagHF method is useful to study the yeast-to-hyphal form transition mechanism in *T. asahii* (14). Magnesium treatment reveals statistical differences in morphological phenotypes between yeast and hyphal forms in *T. asahii*. Organelle morphology also changes with magnesium treatment; it reduces lipid droplet size, increases vacuole size, and restores mitochondrial distribution (14). Large lipid droplets in *T. asahii* grown in Sabouraud broth were similar to those observed when the growth of the red alga *Cyanidioschyzon merolae* stalls under nitrogen deficiency (15) and therefore showed a nutrient-deficient phenotype. A vacuole is a dynamic structure surrounded by a lipid bilayer that relates to osmotic pressure regulation, amino acid stock, and protein degradation caused by autophagy (16, 17). Vacuole enlargement due to DNA damage stress occurs in the hyphae of *Schizosaccharomyces japonicus*, a non-pathogenic fission yeast (18). Vacuolar morphology in *Saccharomyces cerevisiae* is sensitive to extracellular conditions of media: two or three vacuoles form in yeast extract peptone dextrose broth with 100 mM salt; a single large vacuole forms in hypo-osmotic media, and fragmented vacuoles appear in hyper-osmotic media with 1 M salt (16, 19). These studies suggest that organelle shape and size are varied with extracellular conditions, potentially characterizing yeasts at the genus level. The Yeast, A Taxonomic Study, 5th ed. contains information on growth conditions, ubiquinone,

serotyping, arthroconidium, and cell carbohydrates; however, it lacks detailed morphological information (5, 20).

This study aimed to characterize Trichosporonales yeasts by examining magnesium-induced phenotypic changes. Our results revealed the potential benefit of using magnesium treatment for hyphal diversity analyses and a relationship between hyphal morphology and phylogenetic systematics in Trichosporonales.

## RESULTS

### Thirty Trichosporonales yeasts were used in this study

To examine the effects of magnesium sulfate on cell morphologies, 27 Trichosporonales yeasts with sequenced genomes were selected, including species from *Trichosporon* (*T. asahii*, *T. coremiiforme*, *T. faecale*, *T. inkin*, and *T. ovoides*), *Apiotrichum* (*A. brassicae*, *A. domesticum*, *A. gamsii*, *A. gracile*, *A. laibachii*, *A. montevideense*, *A. porosum*, and *A. veenhuisii*), *Cutaneotrichosporon* (*C. arboriforme*, *C. cavernicola*, *C. curvatum*, *C. cutaneum*, *C. cyanovorans*, *C. daszewskae*, *C. dermatis*, *C. mucoides*, and *C. spelunceum*), *Takashimella* (*Ta. koratensis* and *Ta. tepidaria*), *Pascua guehoae*, *Prillingera fragicola*, and *Vanrija humicola* (Table 1). In *Trichosporon*, *T. aquatile*, *T. asteroides*, and *T. japonicum* were added for the analysis. For comparison, non-Trichosporonales hyphal yeasts (*Dipodascus geotrichum*, *Dipodascus reessii*, and *Tausonia pullulans*) and two model yeast forms (*Saccharomyces cerevisiae* and *Schizosaccharomyces pombe*) were used (Table 1).

### Trichosporonales yeast growth was accelerated under magnesium treatment

To examine cell growth influenced by magnesium in each species, $OD_{660}$ values of cells cultivated in standard Sabouraud broth and Sabouraud broth with magnesium sulfate (Sabouraud + Mg) were measured every 10 min for 72 h. Three growth criteria were evaluated: (i) timing of initial $OD_{660}$ increase (growth lag phase), (ii) rate of $OD_{660}$ values per hour, and (iii) $OD_{660}$ values at a growth plateau (Table 2). For the first criterion, 15 species showed an increase in $OD_{660}$ more than 1 h earlier with magnesium treatment compared with that in Sabouraud broth alone (Table 2; Fig. S1 to S3). In contrast, 14 species showed an increase in $OD_{660}$ less than 1 h earlier with magnesium treatment compared with that in Sabouraud broth alone (Table 2; Fig. S1 to S3). The $OD_{660}$ value for *C. daszewskae* did not increase in Sabouraud broth for 72 h (Table 2; Fig. S2). For the second criterion, $OD_{660}$ increase rates accelerated with magnesium treatment across all species. Notably, 23 species displayed more than a twofold increase in the speed of $OD_{660}$ increase under magnesium treatment (Table 2; Fig. S1 to S3). For the third criterion, plateau $OD_{660}$ values increased in all species when treated with magnesium. Twenty-two species showed more than a twofold increase in plateau $OD_{660}$ values under magnesium treatment (Table 2; Fig. S1 to S3). These results suggested that magnesium treatment accelerates growth in Trichosporonales yeasts in Sabouraud broth. A growth phase time point (sampling time) was established for each species to guide further observations (Table 2).

### Cell lengths of 13 species in Trichosporonales were extended with magnesium treatment

To examine cell length extension with magnesium sulfate addition, we measured the cell lengths of 30 Trichosporonales yeasts including 15 hyphal-form species cultivated in Sabouraud broth and Sabouraud + Mg broth: *T. aquatile*, *T. asahii*, *T. asteroides*, *T. coremiiforme*, *T. faecale*, *T. ovoides*, *A. brassicae*, *A. montevideense*, *A. porosum*, *A. laibachii*, *A. veenhuisii*, *C. cutaneum*, *C. cavernicola*, *C. spelunceum*, and *P. guehoae*. Cell length significantly ($P < 0.05$) increased with magnesium treatment in 11 of the 15 hyphal-form species: *T. aquatile* ($P < 10^{-5}$), *T. asahii* ($P < 10^{-5}$), *T. asteroides* ($P < 10^{-5}$), *T. coremiiforme* ($P = 0.002$), *T. ovoides* ($P < 10^{-5}$), *A. brassicae* ($P < 10^{-5}$), *A. montevideense* ($P < 10^{-5}$), *A. veenhuisii* ($P < 10^{-5}$), *C. cavernicola* ($P < 10^{-5}$), *C. cutaneum* ($P = 0.029$), and *P. guehoae* ($P = 0.007$) (Fig. 1 and 2; Table S1). *C. dermatis* showed pseudo-hyphal growth, and its

**TABLE 1** Strains used in the study

| Species | Strains |
|---|---|
| *Trichosporon aquatile* | JCM 3936 |
| *Trichosporon asahii* | JCM 2466 |
| *Trichosporon asteroids* | JCM 2937 |
| *Trichosporon coremiiforme* | JCM 2938 |
| *Trichosporon faecale* | JCM 2941 |
| *Trichosporon inkin* | JCM 9195 |
| *Trichosporon japonicum* | JCM 8357 |
| *Trichosporon ovoides* | JCM 9940 |
| *Apiotrichum brassicae* | JCM 1599 |
| *Apiotrichum domesticum* | JCM 9580 |
| *Apiotrichum gamsii* | JCM 9941 |
| *Apiotrichum gracile* | JCM 10018 |
| *Apiotrichum laibachii* | JCM 2947 |
| *Apiotrichum montevideense* | JCM 9937 |
| *Apiotrichum porosum* | JCM 1458 |
| *Apiotrichum veenhuisii* | JCM 10691 |
| *Cutaneotrichosporon arboriforme* | JCM 14201 |
| *Cutaneotrichosporon cavernicola* | HIS 19 |
| *Cutaneotrichosporon curvatum* | JCM 1532 |
| *Cutaneotrichosporon cutaneum* | JCM 1462 |
| *Cutaneotrichosporon cyanovorans* | JCM 31833 |
| *Cutaneotrichosporon daszewskae* | JCM 11166 |
| *Cutaneotrichosporon dermatis* | JCM 11170 |
| *Cutaneotrichosporon mucoides* | JCM 9939 |
| *Cutaneotrichosporon spelunceum* | HIS 16 |
| *Takashimella koratensis* | JCM 12878 |
| *Takashimella tepidaria* | JCM 11965 |
| *Pascua guehoae* | JCM 10690 |
| *Prillingera fragicola* | JCM 1530 |
| *Vanrija humicola* | JCM 1457 |
| *Dipodascus geotrichum* | JCM 6359 |
| *Dipodascus reessii* | JCM 1943 |
| *Tausonia pullulans* | JCM 9886 |
| *Saccharomyces cerevisiae* | BY1438 |
| *Schizosaccharomyces pombe* | FY7507 |

cell length significantly increased with magnesium treatment ($P = 0.046$). In addition, cell length marginally increased in *Ta. koratensis* ($P < 10^{-5}$), which remained in yeast form regardless of magnesium sulfate addition (Fig. 2; Table S1). Conversely, cell lengths significantly decreased with magnesium treatment in 14 species: *T. faecale* ($P = 0.002$), *T. inkin* ($P < 10^{-5}$), *T. japonicum* ($P = 0.007$), *A. domesticum* ($P < 10^{-5}$), *A. gamsii* ($P < 10^{-5}$), *A. gracile* ($P < 10^{-5}$), *C. arboriforme* ($P < 10^{-5}$), *C. curvatum* ($P < 10^{-5}$), *C. cyanovorans* ($P < 10^{-5}$), *C. daszewskae* ($P < 10^{-5}$), *C. mucoides* ($P < 10^{-5}$), *C. spelunceum* ($P < 10^{-5}$), *Pr. fragicola* ($P < 10^{-5}$), and *Ta. tepidaria* ($P = 0.035$) (Fig. 1 and 2; Table S1). Cell lengths remained statistically unchanged in *A. laibachii*, *A. porosum*, and *V. humicola* upon magnesium sulfate addition (Fig. 1 and 2; Table S1). Therefore, cell lengths in Trichosporonales varied widely in response to magnesium treatment.

## Magnesium treatment increased multi-septation in *Trichosporon* and *C. dermatis*, with no increase in *Apiotrichum*

To examine septation phenotypes in 30 Trichosporonales yeasts with magnesium sulfate treatment, we examined Calcofluor White-stained cellular distributions in cells

**TABLE 2** Comparison of growth among 30 Trichosporonales yeasts in Sabouraud broth[a,b]

| Species | Growth lag phase (h) | Rate of OD$_{660}$ per hour | | | OD$_{660}$ in plateau | | | Sampling time (h) |
|---------|---------------------|------|---------|-------|------|---------|-------|-------------------|
| | | Sab | Sab + Mg | Ratio | Sab | Sab + Mg | Ratio | |
| C. spelunceum | 0.44 | 0.05 | 0.06 | 1.13 | 0.67 | 0.92 | 1.38 | 22 |
| C. cavernicola | 0 | 0.05 | 0.08 | 1.75 | 0.31 | 1.0 | 3.28 | 12 |
| C. cutaneum | 2.0 | 0.03 | 0.05 | 1.43 | 0.72 | 1.39 | 1.92 | 35 |
| C. mucoides | 14.6 | 0.04 | 0.1 | 2.4 | 0.63 | 1.4 | 2.22 | 26 |
| C. dermatis | 2.5 | 0.04 | 0.18 | 4.08 | 0.45 | 1.66 | 3.69 | 16 |
| C. arboriformis | 0 | 0.05 | 0.11 | 2.3 | 0.86 | 1.59 | 1.86 | 16 |
| C. curvatum | 0.11 | 0.02 | 0.13 | 8.2 | 0.3 | 1.81 | 5.94 | 16 |
| C. cyanovorans | 0.06 | 0.05 | 0.13 | 2.62 | 0.69 | 1.86 | 2.68 | 16 |
| P. guehoae | 1.39 | 0.04 | 0.1 | 2.79 | 0.86 | 1.9 | 2.22 | 16 |
| C. daszewskae | NG | NG | 0.06 | ND | 0.03 | 1.08 | 34.5 | 16 |
| Pr. fragicola | 14.4 | 0.03 | 0.12 | 4.55 | 0.37 | 1.62 | 4.37 | 16 |
| V. humicola | −0.72 | 0.08 | 0.19 | 2.38 | 1.47 | 1.81 | 1.24 | 16 |
| A. porosum | 18.9 | 0.02 | 0.11 | 5.5 | 0.56 | 1.96 | 3.49 | 25 |
| A. gamsii | 0 | 0.06 | 0.15 | 2.69 | 0.81 | 1.9 | 2.34 | 16 |
| A. domesticum | 0.56 | 0.04 | 0.21 | 5.82 | 0.41 | 1.95 | 4.81 | 16 |
| A. montevideense | 0 | 0.02 | 0.12 | 7.79 | 0.25 | 1.87 | 7.59 | 16 |
| A. brassicae | 0.22 | 0.04 | 0.12 | 3.28 | 0.63 | 1.31 | 2.07 | 16 |
| A. veenhuisii | 1.83 | 0.01 | 0.17 | 15.6 | 0.15 | 1.45 | 9.63 | 8 |
| A. laibachii | 3.56 | 0.02 | 0.08 | 3.54 | 0.47 | 1.11 | 2.36 | 16 |
| A. gracile | 1.5 | 0.02 | 0.09 | 4.88 | 0.43 | 1.1 | 2.53 | 16 |
| T. inkin | 4.44 | 0.04 | 0.11 | 2.7 | 0.5 | 1.77 | 3.54 | 16 |
| T. ovoides | 5.06 | 0.01 | 0.06 | 8.24 | 0.17 | 1.43 | 8.37 | 16 |
| T. aquatile | 5.11 | 0.06 | 0.11 | 1.97 | 0.8 | 1.3 | 1.62 | 16 |
| T. japonicum | 0.61 | 0.02 | 0.06 | 3.05 | 0.7 | 1.37 | 1.97 | 16 |
| T. asteroides | 33.3 | 0.04 | 0.08 | 1.83 | 1.21 | 1.59 | 1.31 | 27 |
| T. asahii | 12.4 | 0.06 | 0.08 | 1.31 | 0.81 | 1.76 | 2.16 | 16 |
| T. coremiiforme | 0.28 | 0.03 | 0.09 | 3.29 | 0.47 | 1.46 | 3.11 | 16 |
| T. faecale | 6.11 | 0.04 | 0.14 | 3.53 | 0.24 | 1.66 | 6.92 | 16 |
| Ta. tepidarius | 0.22 | 0.04 | 0.08 | 2.18 | 0.61 | 1.24 | 2.06 | 22 |
| Ta. koratensis | 0 | 0.04 | 0.08 | 2.13 | 0.86 | 1.18 | 1.37 | 16 |

[a]Values of OD$_{660}$ increase are shown. Ratio: value of Sab + Mg:value of Sab.
[b]ND, no data; NG, no growth; Sab, Sabouraud broth.

cultivated in Sabouraud broth and Sabouraud + Mg broth (Fig. 3; Fig. S4). Magnesium treatment increased septation frequency in 13 species: *T. aquatile*, *T. asahii*, *T. asteroides*, *T. coremiiforme*, *T. faecale*, *T. ovoides*, *A. brassicae*, *A. domesticum*, *A. gracile*, *A. porosum*, *C. cutaneum*, *C. dermatis*, and *Pr. fragicola* (Fig. 4A and B; Table S1). In addition, magnesium increased multi-septation frequencies in septated cells for seven species: *T. aquatile* (29.8% in Sabouraud broth to 84.5% in Sabouraud + Mg broth), *T. asahii* (32.7%–86.6%), *T. asteroides* (6.5%–22.8%), *T. coremiiforme* (32.4%–73.9%), *T. faecale* (4%–23.6%), *T. ovoides* (0%–58.4%), and *C. dermatis* (18.9%–32%) (Fig. 4A and B; Table S1). Notably, the magnesium treatment impacted multi-septation frequency in *Trichosporon* hyphal forms and in *C. dermatis* pseudo-hyphae (Fig. 3, 4A and B). Therefore, cell length and frequencies of multi-septation cells simultaneously increased in six species: *T. aquatile*, *T. asahii*, *T. asteroides*, *T. coremiiforme*, *T. ovoides*, and *C. dermatis*. Alternatively, magnesium sulfate rarely increased multi-septation frequency in *Apiotrichum* spp. (Fig. 4A and B; Table S1). In *A. brassicae*, *A. montevideense*, and *A. veenhuisii*, multi-septation frequency did not increase with magnesium treatment despite an increase in cell length (Fig. 1 and 4A; Table S1).

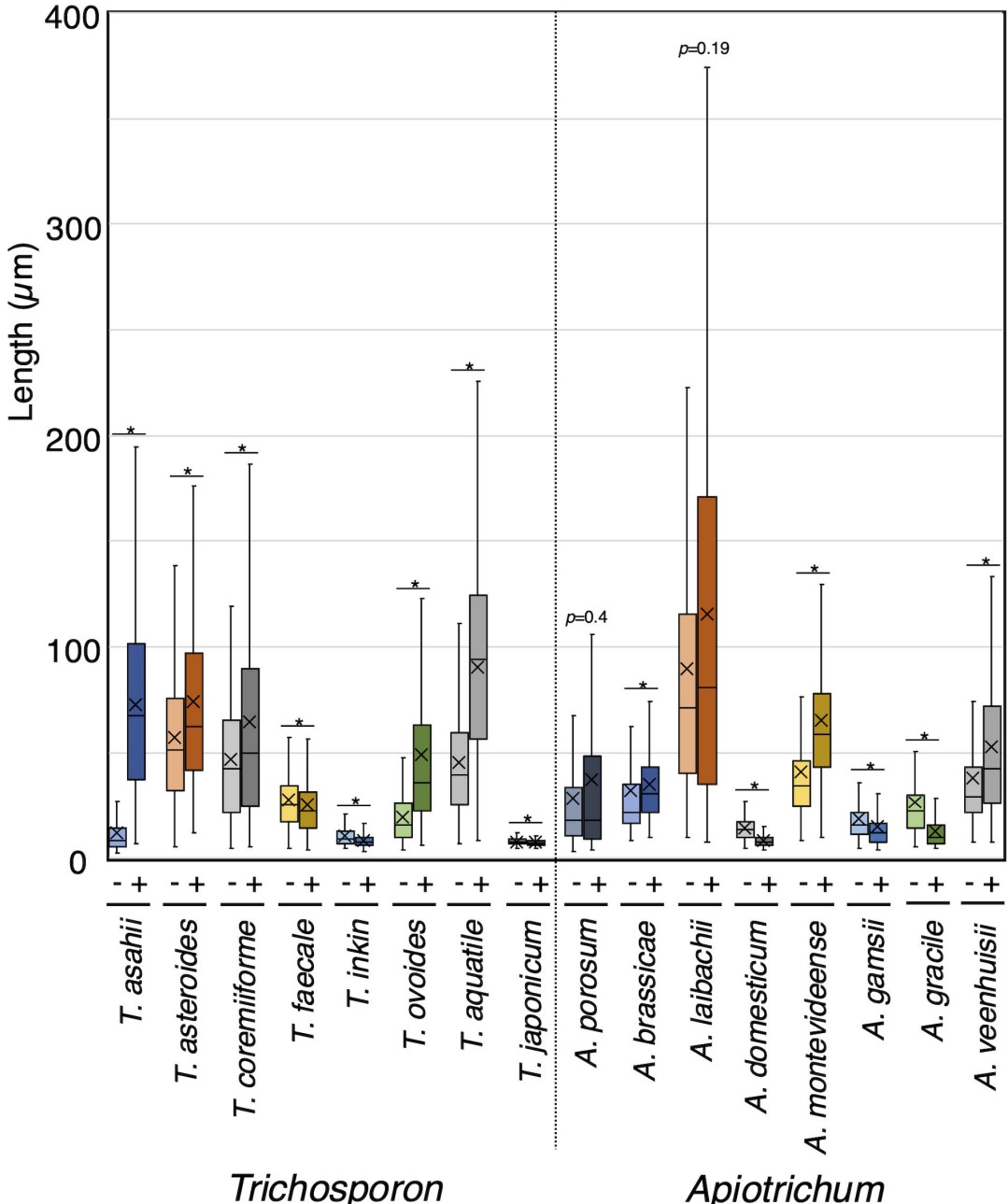

**FIG 1** Cell lengths of *Trichosporon* and *Apiotrichum* with magnesium sulfate treatment. Cell lengths were measured in *Trichosporon* and *Apiotrichum* after cultivation at 25°C in Sabouraud broth (−) and Sabouraud broth supplemented with MgSO$_4$ (+). Each yeast was observed at a species-specific incubation time. Statistical differences between samples were calculated using the Mann–Whitney *U* test, with asterisks indicating significant *P* values (*P* < 0.05). The cross symbols indicate each average length.

## Magnesium treatment impacts lipid droplet size and number

To examine lipid droplet dynamics with magnesium sulfate addition, cells stained with BODIPY were observed under a microscope in 30 Trichosporonales (Fig. 3; Fig. S4). In *T. asahii*, *T. coremiiforme*, and *T. faecale*, lipid droplet size decreased with magnesium treatment (Fig. 3; Fig. S4). Conversely, lipid droplet size increased with magnesium treatment only in *T. inkin* among the 30 Trichosporonales species (Fig. 3; Fig. S4). No phenotypic alterations in lipid droplet size were observed outside *Trichosporon*. In addition, change in lipid droplet number was examined among 30 Trichosporonales

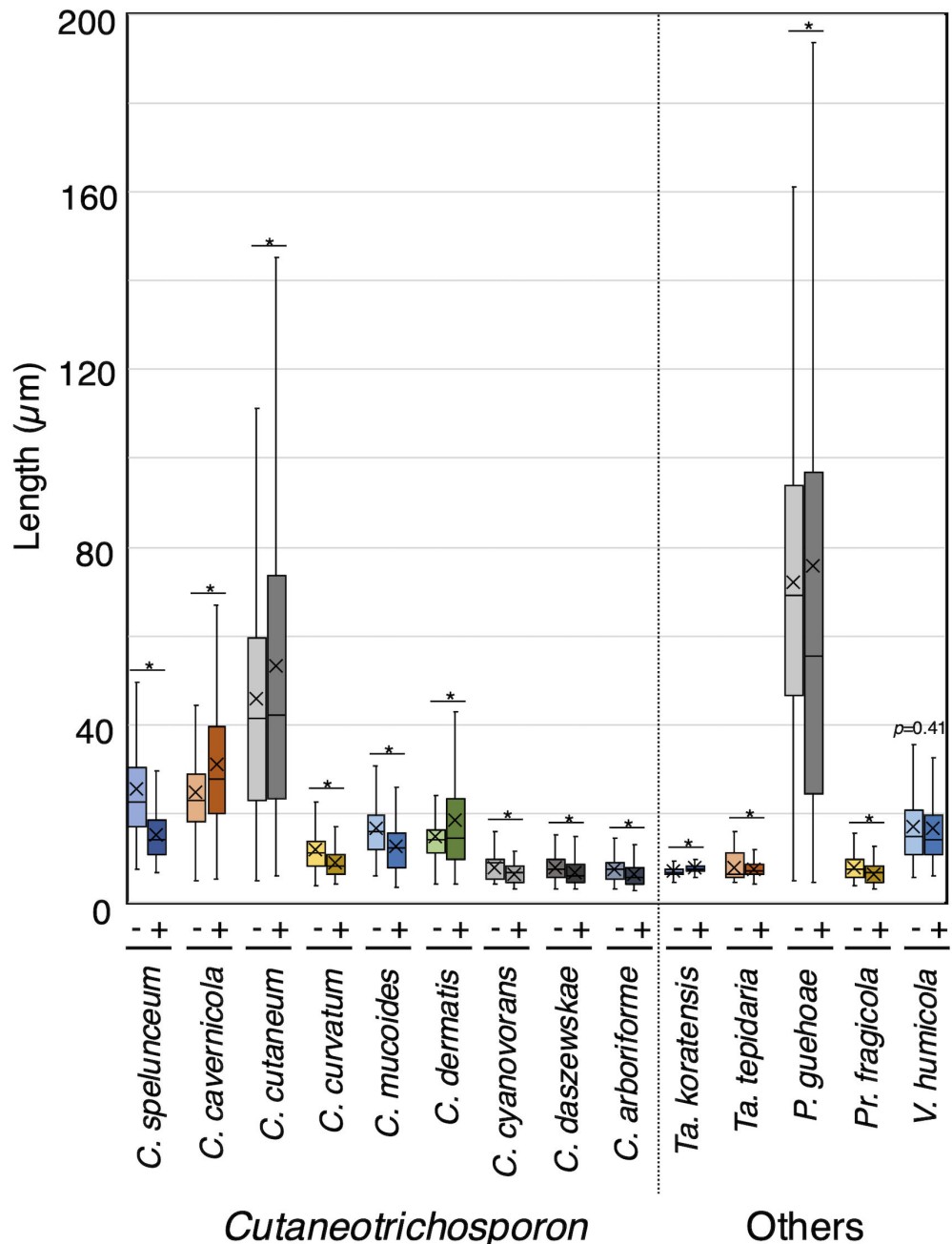

**FIG 2** Cell lengths of *Cutaneotrichosporon* and other Trichosporonales yeasts with magnesium sulfate treatment. Cell lengths were measured in *Cutaneotrichosporon* and other Trichosporonales yeasts after cultivation at 25°C in Sabouraud broth (−) and Sabouraud broth supplemented with $MgSO_4$ (+). Each yeast was observed at a species-specific incubation time. Statistical differences between samples were calculated using the Mann–Whitney $U$ test, with asterisks indicating significant $P$ values ($P$ < 0.05). The cross symbols indicate each average length.

species. Lipid droplet number showed a statistically significant increase with magnesium treatment in 12 species: *T. aquatile* ($P = 0.00038$), *A. brassicae* ($P = 0.00578$), *A. domesticum* ($P = 0.0005$), *A. gamsii* ($P < 10^{-5}$), *A. gracile* ($P = 0.02034$), *A. montevideense* ($P = 0.00006$), *C. arboriforme* ($P = 0.00906$), *C. curvatum* ($P = 0.01108$), *C. cyanovorans* ($P = 0.00222$), *C. dermatis* ($P = 0.00008$), *C. spelunceum* ($P = 0.00034$), and *Ta. tepidaria* ($P < 10^{-5}$) (Fig. 5A and B; Fig. S4; Table S1). In contrast, lipid droplet number showed a statistically significant decrease with magnesium treatment in six species: *T. faecale* ($P = 0.00512$), *T.*

*inkin* ($P = 0.04136$), *C. cavernicola* ($P = 0.03078$), *C. daszewskae* ($P = 0.00782$), *C. mucoides* ($P = 0.0466$), and *P. guehoae* ($P = 0.00124$) (Fig. 5A and B; Fig. S4; Table S1). Decrease in lipid droplet number was not observed in *Apiotrichum* spp.

## Vacuolar extension correlates with increased multi-septation in *T. aquatile*, *T. asahii*, *T. ovoides*, and *C. dermatis*

To examine vacuolar dynamics with magnesium sulfate addition, cells stained with FM4-64, vacuole areas were measured under a microscope in 30 Trichosporonales species (Fig. 3; Fig. S4). Two criteria were used to compare vacuolar areas: the ratio of vacuoles with areas $\geq 10$ $\mu m^2$ and the increase in total vacuolar area per cell. Overall, vacuolar area increased with magnesium treatment in six species: *T. aquatile*, *T. asahii*, *T. inkin*, *T. ovoides*, *C. dermatis*, and *P. guehoae* (Fig. 3, 6 and 7; Table S1). Specifically, the ratios of vacuoles with areas $\geq 10$ $\mu m^2$ significantly increased in *T. asahii* ($P = 0.0011$), *C. dermatis* ($P = 0.0029$), and *P. guehoae* ($P = 0.0409$) (Fig. 6 and 7; Table S1). Total vacuolar area significantly increased in *T. aquatile* ($P = 0.0132$), *T. asahii* ($P = 0.0002$), *T. inkin* ($P = 0.0001$), and *T. ovoides* ($P = 0.0436$) (Fig. 3 and 6; Table S1). *T. asahii* was the only species that showed an increase in both criteria. Conversely, vacuolar areas decreased with magnesium treatment in five species: *A. domesticum*, *A. gracile*, *A. montevideense*, *C. spelunceum*, and *Ta. tepidaria* (Fig. 6 and 7; Table S1). Ratios of vacuoles with areas $\geq 10$ $\mu m^2$ significantly decreased in *A. gracile* ($P = 0.0076$) and *A. montevideense* ($P = 0.0307$) (Fig. 6; Table S1). Moreover, total vacuolar area significantly decreased in *A. domesticum* ($P = 0.0146$), *C. spelunceum* ($P < 10^{-5}$), and *Ta. tepidaria* ($P < 10^{-5}$) (Fig. 6 and 7; Table S1). Therefore, magnesium treatment increased both multi-septation and vacuolar extension phenotypes in *T. aquatile*, *T. asahii*, *T. ovoides*, and *C. dermatis* concurrently. Conversely, these phenotypes did not increase in *A. brassicae*, *A. montevideense*, and *A. veenhuisii*.

## Cellular phenotypes induced by magnesium treatment in non-Trichosporonales yeasts

To study phenotype commonality in cell length, lipid droplet size and number, multi-septation, and vacuolar extension in non-Trichosporonales yeasts, we included five additional yeasts: *D. geotrichum*, *D. reessii*, *T. pullulans*, *S. cerevisiae*, and *Sc. pombe*. Cell lengths significantly increased with magnesium treatment in *D. geotrichum* ($P = 0.042$), *D. reessii* ($P < 10^{-5}$), and *T. pullulans* ($P < 10^{-5}$), which exhibit hyphal forms, and significantly decreased in *S. cerevisiae* ($P < 10^{-5}$) and *Sc. pombe* ($P < 10^{-5}$), which only exhibit yeast forms (Fig. 8A; Table S1). Lipid droplet size was not altered in these yeasts (Fig. S4). However, the lipid droplet number showed a statistically significant increase with magnesium treatment in *D. geotrichum* ($P = 0.00084$), *D. reessii* ($P = 0.0088$), *T. pullulans* ($P = 0.00008$), and *S. cerevisiae* ($P < 10^{-5}$) (Fig. 8B; Table S1; Fig. S4). Multi-septation frequency increased with magnesium treatment in *D. geotrichum* (23.1% in Sabouraud broth to 37.5% in Sabouraud + Mg broth), *D. reessii* (0%–28.6%), and *T. pullulans* (33.3%–72.8%); however, multi-septation was absent in *S. cerevisiae* and *Sc. pombe* (Fig. 8C; Table S1; Fig. S4). Ratios of vacuoles with areas $\geq 10$ $\mu m^2$ significantly increased in *D. reessii* ($P = 0.002$) when magnesium sulfate was added and decreased in *D. geotrichum* ($P = 0.005$) and *T. pullulans* ($P = 0.026$) (Fig. 8D; Table S1; Fig. S4). However, total vacuolar area remained unchanged in the three hyphal yeasts. In *S. cerevisiae*, vacuolar areas were unaffected by magnesium supplementation (Fig. 8D; Table S1; Fig. S4). In *Sc. pombe*, total vacuolar area increased 32-fold with magnesium treatment ($P < 10^{-5}$) (Fig. 8D; Table S1; Fig. S4).

## Phylogenetic tree integrating hyphal phenotypes in Trichosporonales yeasts

To understand the hyphal phenotypes phylogenetically, we constructed a phylogenetic tree using 30 Trichosporonales yeasts, integrating hyphal phenotype information into the tree (Fig. 9). The cell elongation phenotype with magnesium was observed in the *Trichosporon*, *Apiotrichum*, and *Cutaneotrichosporon* (Fig. 9). In contrast, the phenotypes of multi-septation and vacuolar extension were primarily observed in *Trichosporon* (Fig.

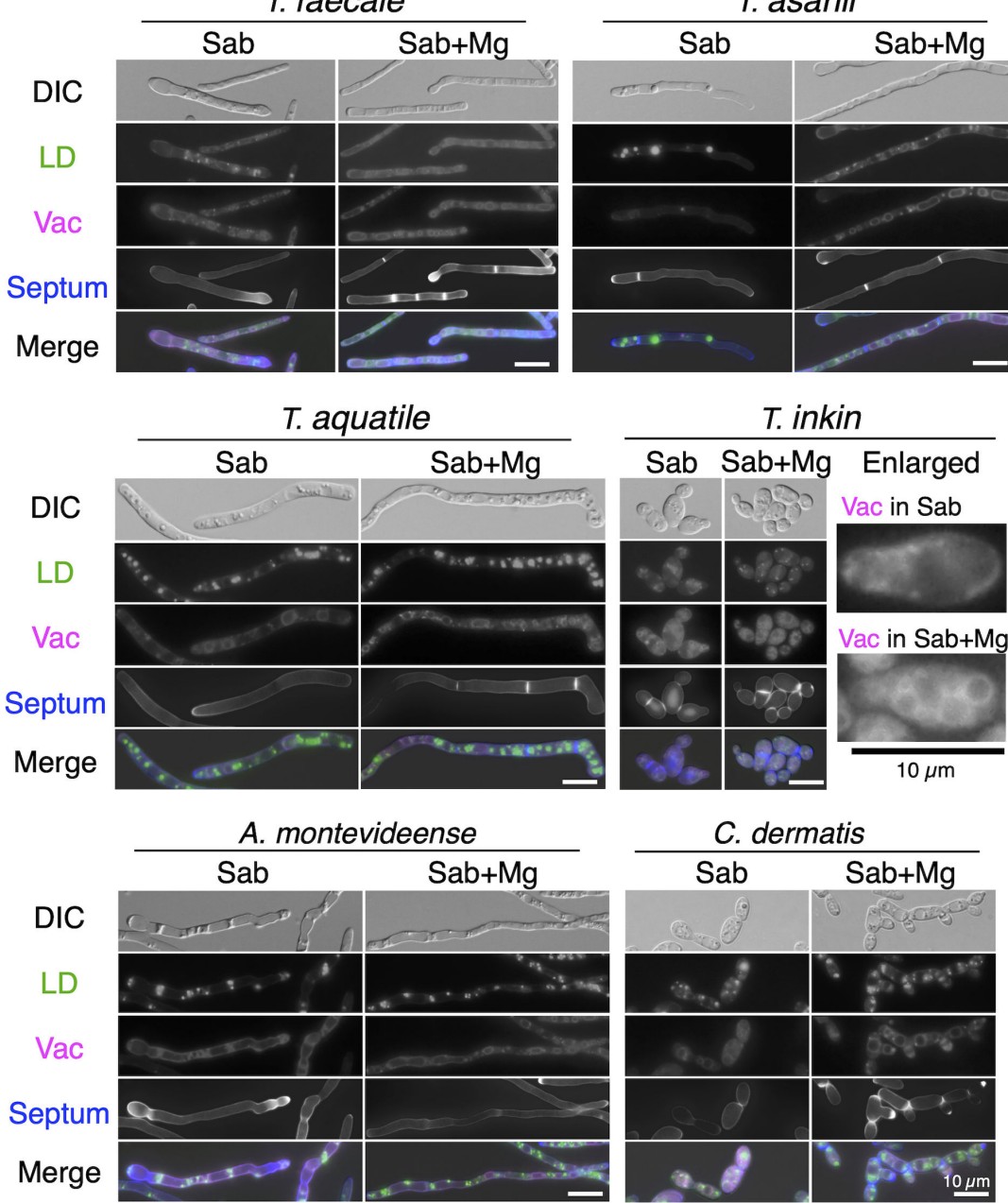

**FIG 3** Cell phenotypes in *T. aquatile*, *T. asahii*, *T. faecale*, *T. inkin*, *A. montevideense*, and *C. dermatis*. Lipid droplets (LDs), vacuoles (Vac), and septum were stained with BODIPY, FM4-64, and Calcofluor White, respectively, using cells cultivated at 25°C in Sabouraud broth (Sab) and Sabouraud broth supplemented with $MgSO_4$ (Sab + Mg) for each species-specific incubation period. *T. inkin* includes enlarged panels stained with FM4-64. Scale bar = 10 µm. DIC, differential interference contrast microscopy.

9). These three phenotypes—cell elongation, multi-septation, and vacuolar extension—concurrently increased with magnesium treatment in *T. aquatile*, *T. asahii*, and *T. ovoides* (Fig. 9). Furthermore, lipid droplet size decreased with magnesium treatment in *T. asahii*, *T. coremiiforme*, and *T. faecale* (Fig. 9). *T. japonicum* did not show changes in any of the four phenotypes (Fig. 9). In addition, cell elongation, multi-septation, and vacuolar extension were all enhanced with magnesium treatment in *C. dermatis* (Fig. 9).

## DISCUSSION

In this study, we examined phenotypic alterations in cell length, lipid droplet size and number, vacuolar area, and multi-septation frequency in Trichosporonales yeasts after magnesium sulfate supplementation in Sabouraud broth. Among the 15 species predominantly forming hyphal structures, 11 species showed a significant extension in cell length following magnesium treatment. The 11 species included 5 from *Trichosporon*, 3 from *Apiotrichum*, 2 from *Cutaneotrichosporon*, and *Pascua humicola*. Notably, the five

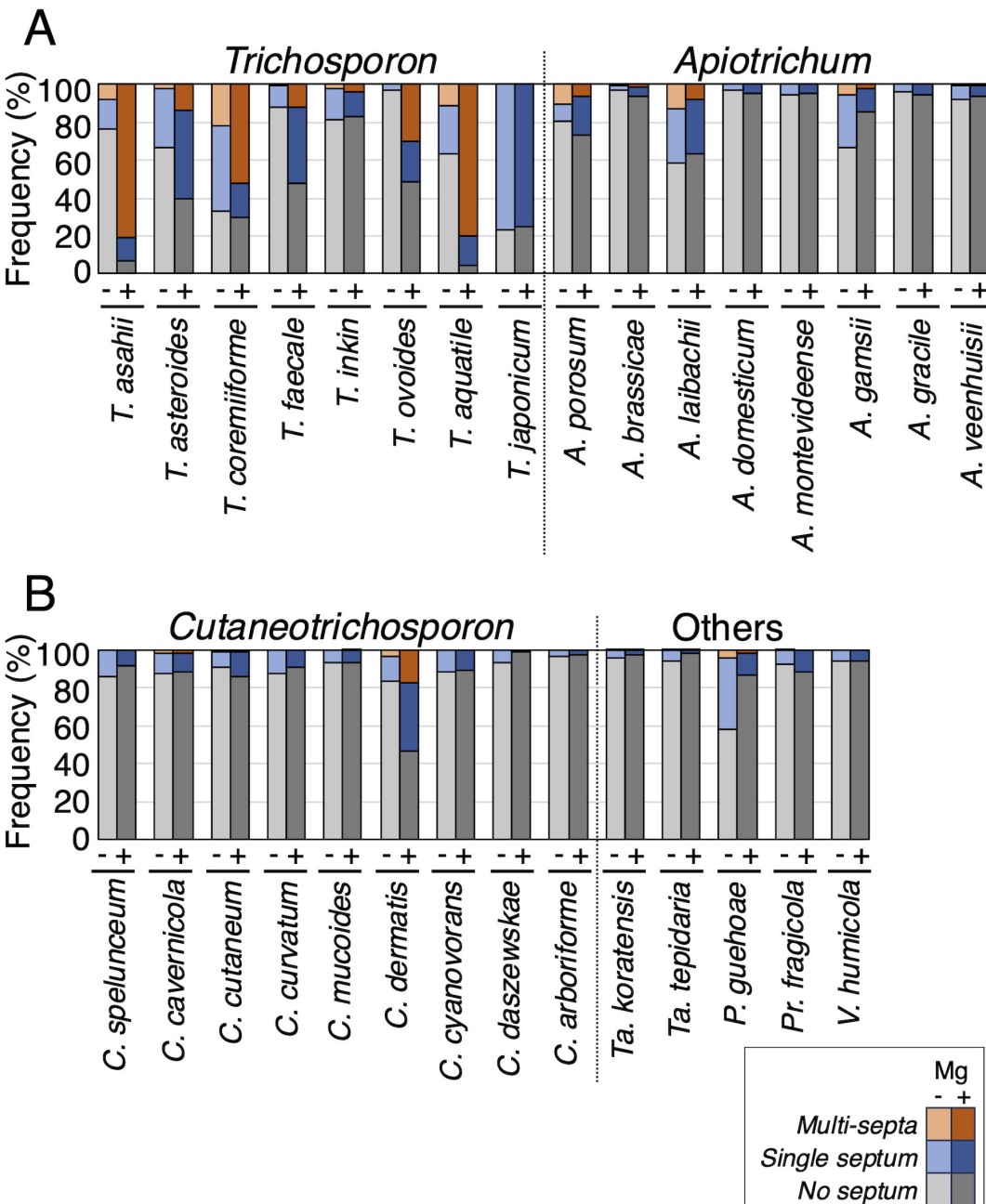

**FIG 4** Frequencies of septation cells in Trichosporonales yeasts. Septum frequencies, stained with Calcofluor White, were measured in (A) *Trichosporon* and *Apiotrichum*, and (B) *Cutaneotrichosporon* and other Trichosporonales yeasts. Measurement was performed after cultivation at 25°C for specified incubation times in Sabouraud broth (−) and Sabouraud broth supplemented with $MgSO_4$ (+). Orange, blue, and gray bars show frequencies of multi-septa, single septum, and no septum, respectively. Light-colored bars correspond to results in Sabouraud broth, and deep-colored bars correspond to results in Sabouraud broth supplemented with $MgSO_4$.

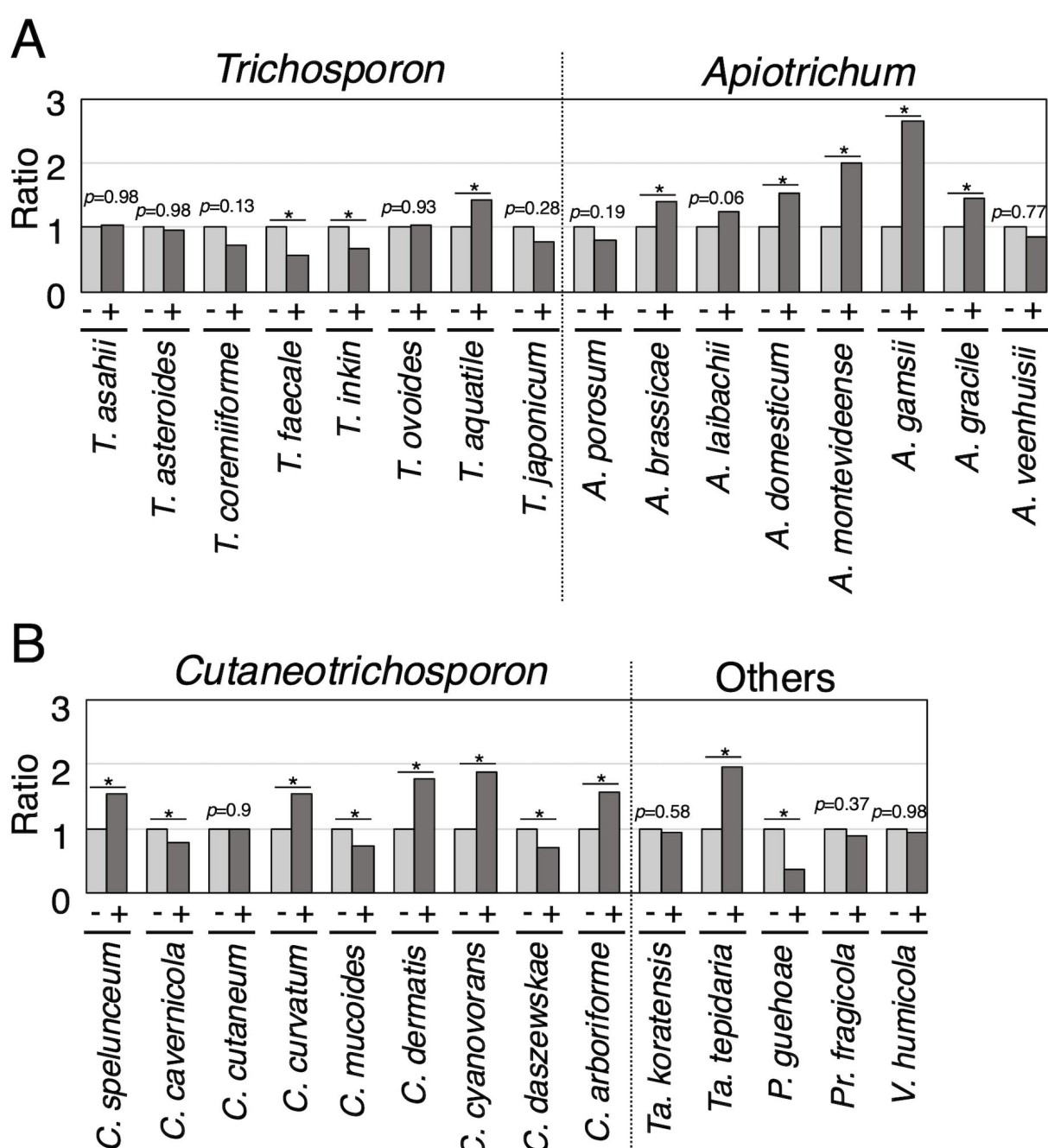

**FIG 5** Comparison of lipid droplet number affected by magnesium treatment. The number of lipid droplets, which were stained with BODIPY, was measured in (A) *Trichosporon* and *Apiotrichum*, (B) *Cutaneotrichosporon*, and other Trichosporonales yeasts. Measurement was performed after cultivation at 25°C for specified incubation times in Sabouraud broth (−) and Sabouraud broth supplemented with $MgSO_4$ (+). Light-gray bars correspond to results for yeasts cultivated in Sabouraud broth, and deep-gray bars correspond to results for yeasts cultivated in Sabouraud broth supplemented with $MgSO_4$. The number of lipid droplets per unit area was calculated. Values for Sabouraud broth supplemented with $MgSO_4$ (+) are expressed relative to those in Sabouraud broth (−). Statistical differences between samples were calculated using the Mann–Whitney $U$ test, with asterisks indicating significant $P$ values ($P < 0.05$).

*Trichosporon* spp., *T. aquatile*, *T. asahii*, *T. asteroides*, *T. coremiiforme*, and *T. ovoides*, showed an increase in multi-septation, with *T. aquatile*, *T. asahii*, and *T. ovoides* also showing vacuolar extension. Vacuolar extension would affect waste product accumulation and protein degradation via autophagy. Conversely, species showing hyphae in *Apiotrichum* and *Cutaneotrichosporon* did not show any increase in the phenotypes of multi-septation

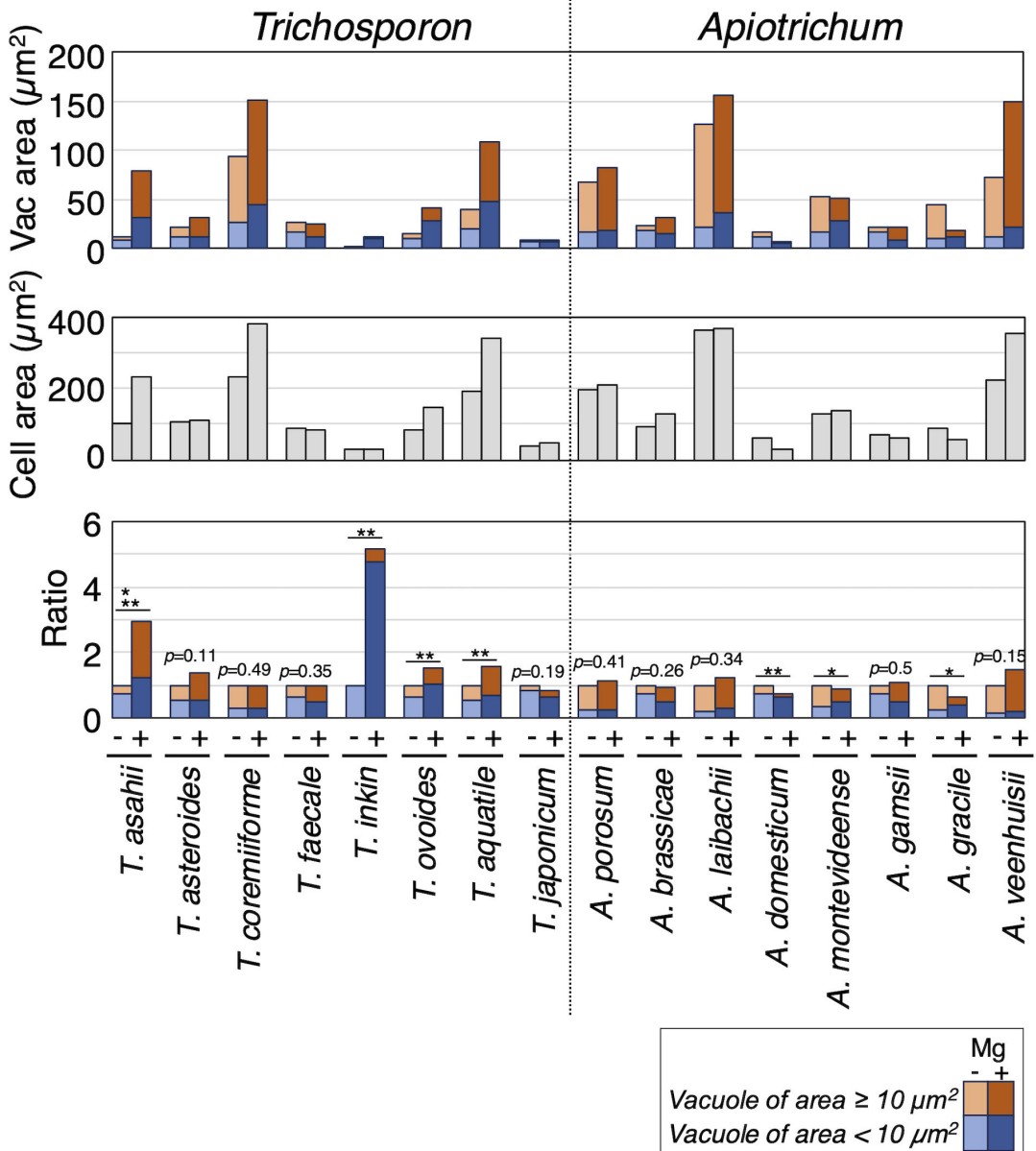

**FIG 6** Comparison of vacuolar area in *Trichosporon* and *Apiotrichum* affected by magnesium treatment. Vacuolar areas stained with FM4-64 were measured in cells cultivated at 25°C for specified times in Sabouraud broth (−) and Sabouraud broth supplemented with $MgSO_4$ (+) in *Trichosporon* and *Apiotrichum*. Measurements were corrected by cell area to calculate the relative ratio of the vacuolar area. Values in Sabouraud broth supplemented with $MgSO_4$ (+) are expressed relative to those in Sabouraud broth (−). Orange, blue, and gray bars are vacuoles with areas ≥10 $\mu m^2$, vacuoles of areas <10 $\mu m^2$, and cell areas, respectively. Light-colored bars represent Sabouraud broth; deep-colored bars represent Sabouraud broth supplemented with $MgSO_4$. Statistical differences between samples were calculated using the Mann–Whitney *U* test. The *P* values indicated by single asterisks and double asterisks are significant at $P < 0.05$. Single asterisks indicate significant differences in vacuoles with areas ≥10 $\mu m^2$. Double asterisks show significant differences in the total vacuolar area. The *P* values in the total vacuolar area are shown when both criteria are not significant.

and vacuolar extension. Our results suggested that *Trichosporon* hyphae exhibit distinct phenotypic features, separate from those of *Apiotrichum* and *Cutaneotrichosporon*, particularly when magnesium is added and support their phylogenetic separation based on these traits at the genus level. The phenotypic features of *Trichosporon* might be related to the features of serotype II because *Trichosporon* is only classified into serotype II and distinguished from *Apiotrichum* and *Cutaneotrichosporon* (4). *T. japonicum*, which is

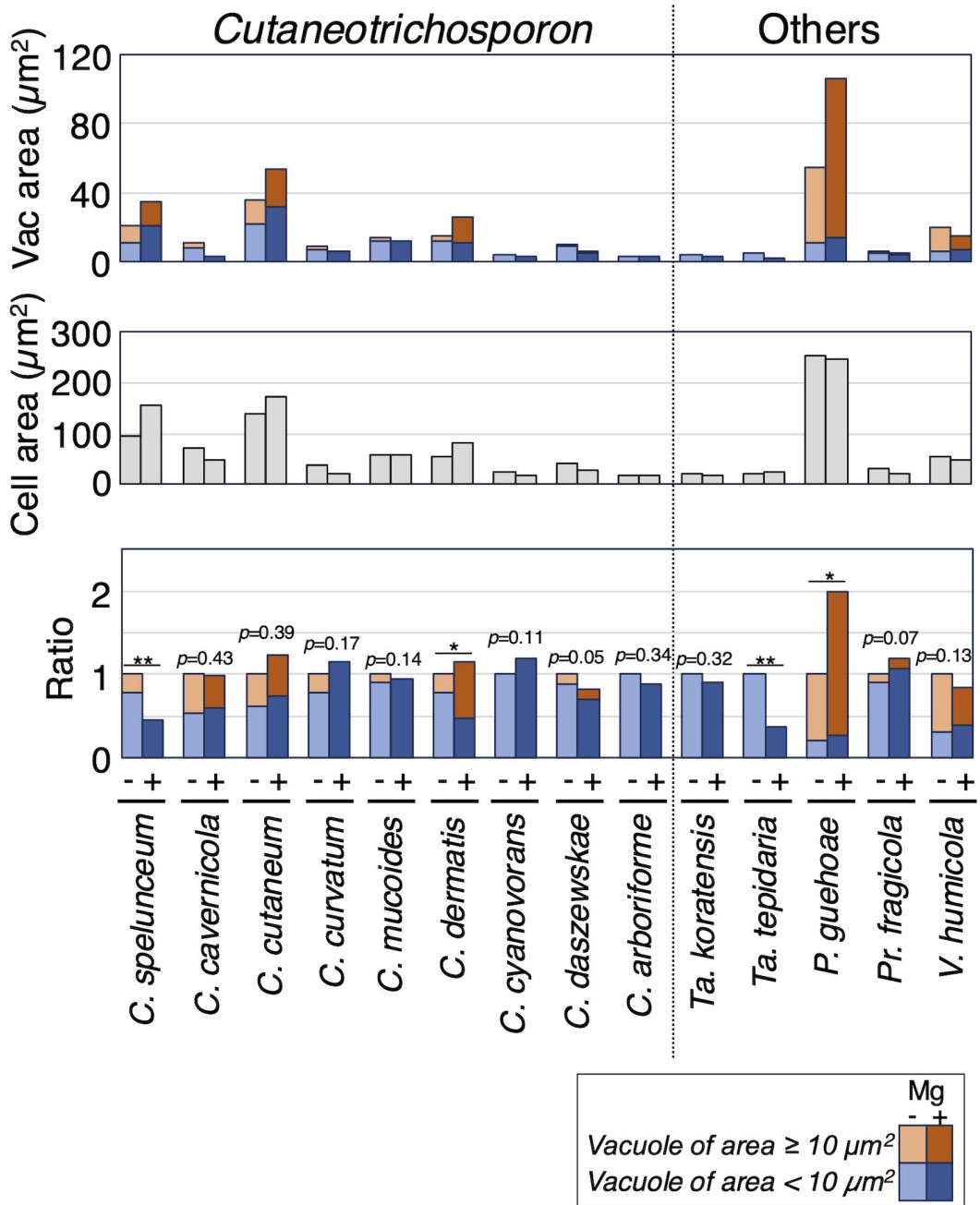

**FIG 7** Comparison of vacuolar area in *Cutaneotrichosporon* and other Trichosporonales yeasts affected by magnesium treatment. Vacuolar areas stained with FM4-64 were measured in cells cultivated at 25°C for specified times in Sabouraud broth (−) and Sabouraud broth supplemented with $MgSO_4$ (+) in *Cutaneotrichosporon*, *Takashimella*, *P. guehoae*, *Pr. fragicola*, and *V. humicola*. Measurements were corrected by the cell area to calculate the relative ratio of the vacuolar area. Values in Sabouraud broth supplemented with $MgSO_4$ (+) are expressed relative to those in Sabouraud broth (−). Orange, blue, and gray bars are vacuoles with areas ≥10 $\mu m^2$, vacuoles of area <10 $\mu m^2$, and cell areas, respectively. Light-colored bars represent Sabouraud broth; deep-colored bars represent Sabouraud broth supplemented with $MgSO_4$. Statistical differences between samples were calculated using the Mann–Whitney *U* test. The *P* values indicated by single asterisks and double asterisks are significant at *P* < 0.05. Single asterisks indicate significant differences in vacuoles with areas ≥10 $\mu m^2$. Double asterisks show significant differences in the total vacuolar area. The *P* values in the total vacuolar area are shown when both criteria are not significant.

known to form appressoria on corn meal agar (21), is a sticky yeast-form exception in the *Trichosporon* genus.

A change in lipid droplet size was observed only in *Trichosporon*. *T. inkin* showed a notable increase in lipid droplet size as a result of magnesium treatment among the

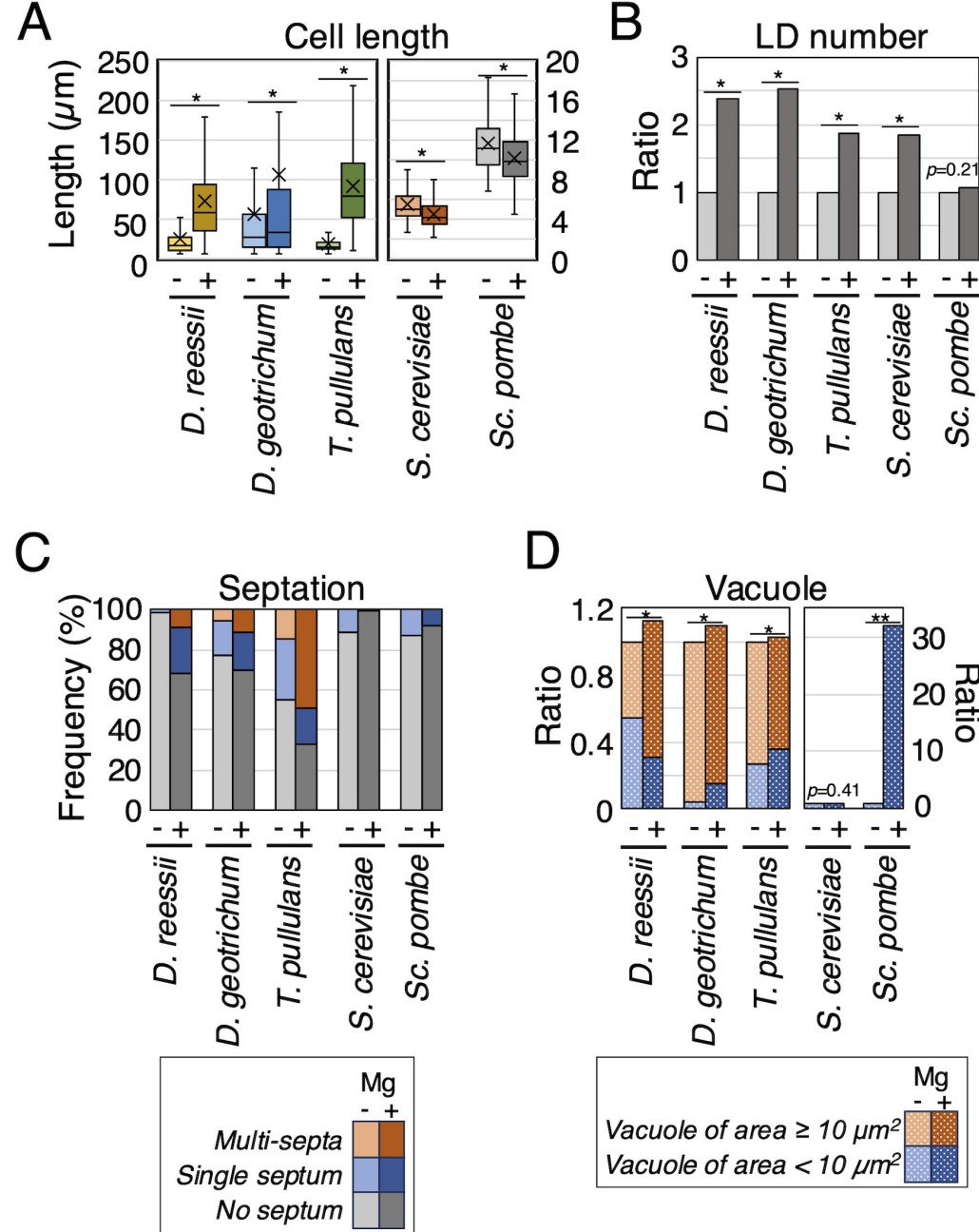

**FIG 8** Phenotypes of cell length, septation, lipid droplet, and vacuole in non-Trichosporonales. Cell lengths, lipid droplet number, septation cells, and vacuolar areas were measured in non-Trichosporonales yeasts: *Dipodascus geotrichum*, *Dipodascus reessii*, *Tausonia pullulans*, *Saccharomyces cerevisiae*, and *Schizosaccharomyces pombe*. Cells were cultivated for each at 25°C for specified incubation times in Sabouraud broth (−) and Sabouraud broth supplemented with $MgSO_4$ (+) for specified incubation times. (A) Cell lengths were measured in the five non-Trichosporonales yeasts. Asterisks indicate *P* values significant at *P* < 0.05. Cross marks represent each average length. (B) Lipid droplet numbers were measured in the five non-Trichosporonales yeasts. Lipid droplet numbers per unit area were calculated. Values in Sabouraud broth supplemented with $MgSO_4$ (+) are expressed relative to those in Sabouraud broth (−). Asterisks indicate *P* values significant at *P* < 0.05. (C) Septum frequencies were measured. Orange, blue, and gray bars show frequencies of multi-septa, single septum, and no septum, respectively. Light-colored bars correspond to results in Sabouraud broth, and deep-colored bars correspond to results in Sabouraud broth supplemented with $MgSO_4$. (D) Vacuolar areas were measured. Measurements were corrected by the cell area to calculate the relative ratio of the vacuolar area. Values for Sabouraud broth supplemented with $MgSO_4$ (+) are expressed relative to those in Sabouraud broth (−). Orange and blue meshed bars correspond to vacuoles

Fig 8 (Continued)

with areas ≥10 $\mu m^2$ and vacuoles of areas <10 $\mu m^2$, respectively. Light-colored meshed bars represent Sabouraud broth, and deep-colored meshed bars represent Sabouraud broth supplemented with $MgSO_4$. Single asterisks indicate significant difference in vacuoles with areas ≥10 $\mu m^2$. A double asterisk indicates a significant difference in total vacuolar area.

30 Trichosporonales yeasts. *T. coremiiforme* and *T. faecale* accumulated lipid droplets in Sabouraud broth as *T. asahii* did. Therefore, *T. asahii*, *T. coremiiforme*, and *T. faecale* show a common nutrient-deficient phenotype in Sabouraud broth. The accumulated lipid droplets decreased in size under magnesium treatment in the three hyphae. Nutrients may be released from shrunken lipid droplets via magnesium treatment, increasing their supply to mitochondria. This may promote hyphal growth in *T. asahii* and *T. coremiiforme*. The decrease in lipid droplet size distinguished *T. asahii*, *T. coremiiforme*, and *T. faecale* hyphae from *T. aquatile*, *T. asteroides*, and *T. ovoides* hyphae within *Trichosporon*. Therefore, it is suggested that the lipid droplet size phenotype induced by magnesium treatment distinguishes yeast species in *Trichosporon*. In addition, *C. dermatis* forms pseudo-hyphae in the broth and shows increased multi-septation and vacuolar area with magnesium treatment, unlike other *Cutaneotrichosporon* spp. This suggests that *C. dermatis* is clearly distinguished from *Cutaneotrichosporon* on the phylogenetic tree at the species level and might relate to the previous report that *C. dermatis* is the only yeast included in the SHP-associated yeasts among *Cutaneotrichosporon* spp. used in the study (6).

A change in lipid droplet number was observed across *Trichosporon*, *Apiotrichum*, and *Cutaneotrichosporon*, independent of their morphologies. Contrary to the case of lipid droplet size, the lipid droplet number was preferentially altered via magnesium treatment in *Apiotrichum* and *Cutaneotrichosporon*. Therefore, the phenotype of lipid droplet size and number via magnesium treatment may distinguish *Trichosporon* from *Apiotrichum* and *Cutaneotrichosporon* in Trichosporonales yeasts.

Some hyphae showed unique phenotypes among the 30 Trichosporonales yeasts. *T. faecale* and *P. guehoae* are exceptional because *T. faecale* cells increased multi-septation without cell length extension, and *P. guehoae* cells increased cell length and vacuolar area without an increase in multi-septation. The cell length extension and multi-septation phenotypes in response to magnesium treatment were observed in the hyphae of *D. geotrichum*, *D. reessii*, and *T. pullulans*. These results suggest that the nature of cell extension induced via magnesium treatment is common across species other than Trichosporonales, including ascomycetous and basidiomycetous yeasts.

Magnesium was confirmed to be an essential nutrient for the growth of all yeasts tested in the study, as evidenced by increased $OD_{660}$ values across forms. Among 13 species dominated by yeast form in broth, 11 species showed statistically significant decreases in cell length under magnesium treatment. This decrease may result from cell shrinkage due to dehydration from the high osmolarity of the magnesium treatment, as observed in *S. cerevisiae* (22). *Ta. koratensis* cells were an exception, showing slight extension with magnesium treatment, possibly due to higher resistance to osmolarity. In three species, *A. domesticum*, *A. gracile*, and *Ta. tepidaria*, the vacuolar area significantly decreased with magnesium treatment. Decreases in vacuolar area may also be impacted by high osmolarity of the magnesium treatment in the broth. *T. inkin* is the only yeast that decreases cell length and increases lipid droplet size and vacuolar area with magnesium treatment. In *T. inkin*, total vacuolar area increased approximately fivefold with magnesium treatment, even without cell area expansion, indicating an increase in vacuole number. The vacuolar phenotype of *T. inkin* was similar to that of *Sc. pombe*. Vacuolar formation in *T. inkin* and *Sc. pombe* may need magnesium uptake in Sabouraud broth, differing from the response observed in *S. cerevisiae*. Overall, yeast forms in Trichosporonales would lack hyphal formation mechanisms; therefore, the phenotype of dehydration would appear.

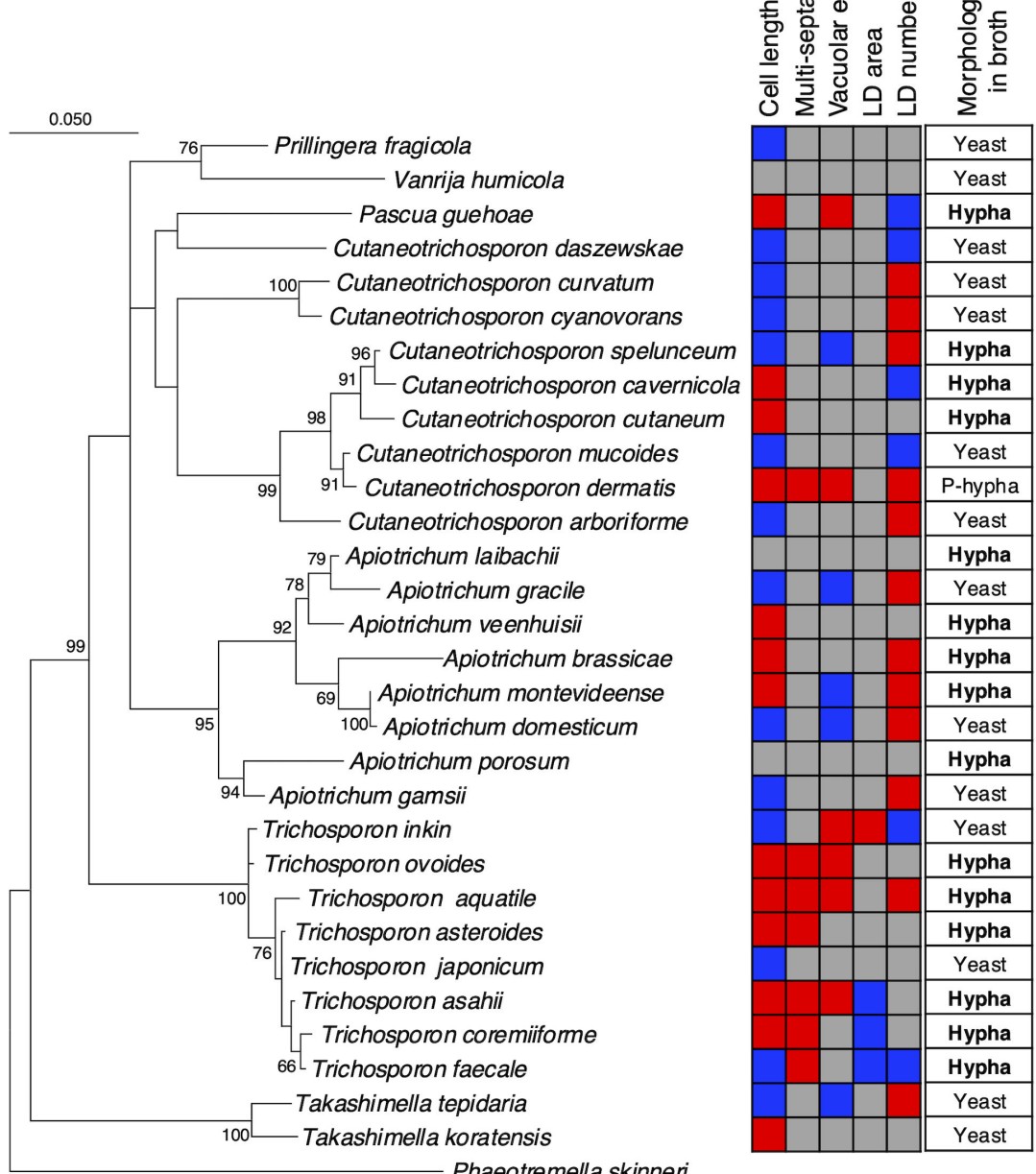

**FIG 9** Phylogenetic tree integrated with hyphal phenotypes. A phylogenetic tree was described using 30 Trichosporonales yeasts. *Phaeotremella skinneri* was used as the outgroup. Six rows indicating cell length, multi-septa, vacuolar extension, lipid droplet (LD) area, LD number, and morphology in broth are described. The red column indicates "increase"; the blue column indicates "decrease"; and the gray column indicates "not impacted" due to magnesium treatment. Morphologies during the growth phase were characterized as yeast, hypha, and pseudo-hypha (P-hypha).

Our results suggest that magnesium-induced morphological changes may contribute to the classification of hyphal forms in Trichosporonales yeasts. The MagHF method using magnesium treatment (14) would be useful for classifying hyphae across species in ascomycetous and basidiomycetous yeasts. Examining the effects of magnesium on morphological phenotypes of additional organelles such as the nucleus, mitochondria, and endoplasmic reticulum could further aid in characterizing Trichosporonales yeasts. This study provides a basis for understanding fungal pathologies and may inform future treatment developments. However, there are limitations to the study that need to be overcome in further studies to fully understand how cells elongate via magnesium

addition. Further studies should aim to analyze the expression of genes induced by magnesium treatment.

## MATERIALS AND METHODS

### Strains and media

Strains were provided by the Japan Collection of Microorganisms (RIKEN BioResource Research Center, Tsukuba, Japan), with the exception of *S. cerevisiae* and *Sc. pombe*. *S. cerevisiae* and *Sc. pombe* were provided by the National Bio-Resource Project, Japan. Strains used in the study are listed in Table 1. These strains were cultivated in Sabouraud broth (4% glucose and 1% bacto peptone) with or without 4.15 mM $MgSO_4$.

### Cell length measurement

The study procedure was primarily based on the MagHF method (14), with minor modifications. Cells ($OD_{660}$ = 0.1) were inoculated in 5 mL Sabouraud broth with or without 4.15 mM $MgSO_4$. Cultures were incubated in a BioShaker BR-23FP air shaker (TAITEC, Koshigaya City, Japan) at 25°C and 100 rpm until the growth phase was reached. The appropriate incubation time of each species is listed as sampling times in Table 2. $OD_{660}$ values were measured using mini photo 518R (TAITEC). After incubation, each sample (1 mL) was transferred to a 1.5 mL tube and centrifuged at 2,260 × $g$. The supernatant was removed, leaving 20 µL of the medium in each sample tube. The cell sample was mixed via tapping and observed under an OLYMPUS BX53 microscope at ×40 or ×100 magnification with a U-FDICT mirror unit (Olympus, Tokyo, Japan). Cell length was measured between cell tips along the cell shape using CellSens software (Olympus). The *P* value for each cell length was estimated using the Mann–Whitney *U* test. A *P* value of <0.05 was considered statistically significant.

### Growth rate measurement

The procedure followed was based on a previous study by Aoki et al. (14). Cells were inoculated into 5 mL of each liquid medium when the $OD_{660}$ value was 0.05. Samples were incubated in a Bio-photorecorder TVS062CA air shaker (ADVANTEC) at 100 rpm for 46 h (shortest time) to 72 h (longest time) at 25°C. The $OD_{660}$ values were recorded every 10 min.

### Cell staining

The procedure followed that of Aoki et al. (14). After the incubation of 5 mL cultures, 1 mL of cell culture was harvested and washed once with 1 mL phosphate-buffered saline (PBS). Thereafter, the cells were mixed with 1 mL PBS containing 100 nM BODIPY (Thermo Fisher Scientific, Waltham, MA, USA) and 2 mM FM4-64 (1 µL; TOCRIS, Bristol, UK) and left in the dark for 5 min at 25°C–28°C. The optimal incubation time for BODIPY staining in live cells for 0, 1, 2, 5, and 10 min was examined. BODIPY signals were bright enough even after 1 min incubation, and the brightness of BODIPY signals did not differ among incubation times from 1 to 10 min (Fig. S5). For cell wall staining, Calcofluor White solution (1 µL; Sigma-Aldrich, St. Louis, USA) was added after being diluted 1,000-fold. After washing cells once with PBS (1 mL), they were resuspended in 50 µL PBS before observation. BODIPY was diluted to a concentration of 100 µM in dimethyl sulfoxide (DMSO) and stored at −20°C. FM4-64 was diluted to 10 mM in DMSO and stored at −20°C. The U-FGFP, U-FGW, and U-FUNA mirror units were used for observing BODIPY, FM4-64, and Calcofluor White signals, respectively (Olympus). Light was supplied using the U-HGLGPS (Olympus).

## Septum number measurement

The number of septa that were stained with Calcofluor White was measured using CellSens software (Olympus). Cells were classified into three groups: multi-septa, single septum, and no septum. Cells with two or more septa were considered multi-septa.

## Quantification of vacuolar area

The area of vacuoles that were stained with FM4-64 was measured using CellSens software (Olympus). Vacuoles in cells were classified into two different size groups: ≥10 and <10 $\mu m^2$. The average area of each group was calculated per cell. The average vacuolar area values were corrected by the average values of the cellular area. The corrected value area in Sabouraud + Mg broth (+) is shown as a relative value in the graph compared to the corrected value in Sabouraud broth (−).

## Phylogenetic tree

In total, 31 D1/D2 and internal transcribed spacer plus 5.8S rRNA sequences were collected from the database and separately aligned using MAFFT (DOI: 10.1093/molbev/mst010), followed by manual refinement. The two trimmed alignments were concatenated into one, resulting in a combined alignment of 951 nucleic acid sites. The evolutionary history was inferred using the maximum likelihood method based on the Tamura–Nei model (23). The tree with the highest log likelihood (−5,265.93) is shown. The percentage of trees where associated taxa clustered together is shown next to the branches. Initial trees for the heuristic search were obtained automatically by applying the maximum parsimony method. A discrete gamma distribution was used to model evolutionary rate differences among sites (five categories; +G, parameter = 0.5567). Some sites were modeled as evolutionarily invariable ([+I], 50.24% sites). The tree is drawn to scale, with branch lengths measured by the number of substitutions per site. The analysis included 31 nucleotide sequences, eliminating positions containing gaps or missing data, resulting in 951 positions in the final data set. Evolutionary analyses were performed using MEGA7 (24).

## Statistical analysis

Statistical differences between samples were calculated using the Mann–Whitney $U$ test with two-tailed hypothesis. Each data set was introduced into the calculator on the website (https://www.socscistatistics.com/tests/mannwhitney/default2.aspx), and the statistical significance for each was calculated. The $P$ values are significant at $P < 0.05$.

## ACKNOWLEDGMENTS

This work was supported by the Institute for Fermentation, Osaka, Japan. We thank Editage (www.editage.com) for English language editing.

## AUTHOR AFFILIATIONS

[1]Laboratory of Yeast Systematics, Tokyo NODAI Research Institute, Tokyo University of Agriculture, Setagaya, Tokyo, Japan
[2]Japan Collection of Microorganisms, RIKEN BioResource Research Center, Tsukuba, Ibaraki, Japan
[3]Department of Microbiology, Meiji Pharmaceutical University, Kiyose, Tokyo, Japan
[4]Department of Molecular Microbiology, Faculty of Life Sciences, Tokyo University of Agriculture, Setagaya, Tokyo, Japan

## AUTHOR ORCIDs

Keita Aoki http://orcid.org/0000-0003-2079-4031
Takashi Sugita http://orcid.org/0000-0002-2127-5017

Masako Takashima ⬤ http://orcid.org/0000-0002-7686-8661

## FUNDING

| Funder | Grant(s) | Author(s) |
| --- | --- | --- |
| Institute for Fermentation, Osaka (IFO) | | Masako Takashima |

## AUTHOR CONTRIBUTIONS

Keita Aoki, Conceptualization, Data curation, Formal analysis, Investigation, Methodology, Project administration, Validation, Visualization, Writing – original draft | Moriya Ohkuma, Resources, Writing – review and editing | Takashi Sugita, Writing – review and editing | Yuuki Kobayashi, Methodology, Writing – review and editing | Naoto Tanaka, Supervision, Writing – review and editing | Masako Takashima, Resources, Supervision, Writing – review and editing

## ADDITIONAL FILES

The following material is available online.

### Supplemental Material

**Supplemental material (Spectrum03210-24-s0001.pdf).** Additional experimental details.

### Open Peer Review

**PEER REVIEW HISTORY (review-history.pdf).** An accounting of the reviewer comments and feedback.

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
