## [Reviewer comments · Microbiology Spectrum]

Microbiology Spectrum

Analyses of hyphal diversity in Trichosporonales yeasts based on fluorescent microscopic observations

Keita Aoki, Moriya Ohkuma, Takashi Sugita, Yuuki Kobayashi, Naoto Tanaka, and Masako Takashima

Corresponding Author(s): Keita Aoki, Tokyo Nogyo Daigaku

Review Timeline:

Submission Date:	December 8, 2024
Editorial Decision:	January 9, 2025
Revision Received:	January 31, 2025
Accepted:	February 9, 2025

Editor: Michael Klutstein

Reviewer(s): Disclosure of reviewer identity is with reference to reviewer comments included in decision letter(s). The following individuals involved in review of your submission have agreed to reveal their identity: Wencheng Zhu (Reviewer #1); MARWAN Yaseen AL-MAQTOOFI (Reviewer #2)

Transaction Report:

DOI: <https://doi.org/10.1128/spectrum.03210-24>

Re: Spectrum03210-24 (Analyses of hyphal diversity in Trichosporonales yeasts based on fluorescent microscopic observations)

Dear Dr. Keita Aoki:

Thank you for the privilege of reviewing your work. Below you will find my comments, instructions from the Spectrum editorial office, and the reviewer comments.

Revision Guidelines

Sincerely,
Michael Klutstein
Editor
Microbiology Spectrum

Reviewer #1 (Comments for the Author):

In this study, the authors systematically analyzed phenotypic changes in Trichosporonales yeasts following magnesium treatment. The study design is simple, but the results are informative. This manuscript is recommended for publication after addressing the following issues.

1. LD numbers should be calculated. In species such as *Trichosporon aquatile*, *Apiotrichum brassicae*, and *Takashimella tepidaria*, although the LD area did not show significant changes before and after magnesium treatment, the number of LDs increased following the treatment.
2. In species like *Trichosporon asahii* and *Trichosporon faecale*, magnesium treatment significantly decreased the number of LDs, which may indicate a shift in cellular metabolism. I recommend that the authors to discuss about this observation to explore its biological significance.
3. In figures 4-8, the authors should directly indicate what the colors represent in the figure itself, rather than only in the figure legend. This will help clarify the interpretation of the data for readers.
4. A helpful tip for future studies would be for the authors to test the incubation time with BODIPY when staining LDs to optimize the staining conditions for the best results.

Reviewer #2 (Comments for the Author):

I would prefer if the author can add a special section for statistical analysis.

January 31, 2025

Prof. Michael Klutstein

Editor

Microbiology Spectrum

Dear Prof. Klutstein,

These are our point-by-point responses to the comments and suggestions raised by Reviewer 1 and Reviewer 2. Bold-type text indicates the comments from the Reviewers.

To Reviewer 1

Reviewer 1 (Comments for the Author):

In this study, the authors systematically analyzed phenotypic changes in Trichosporonales yeasts following magnesium treatment. The study design is simple, but the results are informative. This manuscript is recommended for publication after addressing the following issues.

1. LD numbers should be calculated. In species such as *Trichosporon aquatile*, *Apiotrichum brassicae*, and *Takashimella tepidaria*, although the LD area did not show significant changes before and after magnesium treatment, the number of LDs increased following the treatment.

- We appreciate the suggestion of this new approach for analyzing phenotypes. We measured lipid droplet numbers in the cells that were used for examining vacuolar area in Fig. 6, 7, and 8 in the study. Thus, cellular areas, which is necessary for calculating values per unit area, were shared with the measurement of vacuolar areas.

As a result, lipid droplet number statistically increased with magnesium treatment in twelve species: *T. aquatile*, *A. brassicae*, *A. domesticum*, *A. gamsii*, *A. gracile*, *A. montevidense*, *C. arboriforme*, *C. curvatum*, *C. cyanovorans*, *C. dermatis*, *C. spelunceum*, and *Ta. tepidaria*. In contrast, lipid droplet number statistically decreased with magnesium treatment in six species: *T. faecale*, *T. inkin*, *C. cavernicola*, *C. daszewskae*, *C. mucoides*, and *P. guehoae*. Decrease in lipid droplet number was not observed in

Apiotrichum species. In non-Trichosporonales yeasts, lipid droplet number statistically increased with magnesium treatment in *D. geotrichum*, *D. reessii*, *T. pullulans*, and *S. cerevisiae*. Details regarding these results were added on Page 11 line 19, and Page 13 line 17, respectively. As reviewer 1 has kindly mentioned, lipid droplet numbers increased following magnesium treatment in *T. aquatile*, *A. brassicae*, and *Ta. tepidaria*.

The graphs depicting the change in lipid droplet numbers with magnesium treatment were newly described in Fig. 5 and 8B. The legend of Fig. 5 and 8B were added in Page 28 line 14, and Page 30 line 22, respectively. In addition, the row of lipid droplet numbers was added to the phylogenetic tree depicted in Fig. 9.

We discussed the relationship between lipid droplet size and number on Page 16 line 17 as follows; ‘A change in lipid droplet number was observed across *Trichosporon*, *Apiotrichum*, and *Cutaneotrichosporon*, independent of their morphologies. Contrary to the case of lipid droplet size, lipid droplet number was preferentially altered via magnesium treatment in *Apiotrichum* and *Cutaneotrichosporon*. Therefore, the phenotype of lipid droplet size and number via magnesium treatment may distinguish *Trichosporon* from *Apiotrichum* and *Cutaneotrichosporon* in Trichosporonales yeasts.’

2. In species like *Trichosporon asahii* and *Trichosporon faecale*, magnesium treatment significantly decreased the number of LDs, which may indicate a shift in cellular metabolism. I recommend that the authors to discuss about this observation to explore its biological significance.

- Measurement of lipid droplet number in the revision revealed that lipid droplet number significantly decreased via magnesium treatment in *T. faecale*, but not in *T. asahii*, whereas lipid droplet size significantly decreased via magnesium treatment in *T. asahii*, *T. coremiiforme*, and *T. faecale*. We added a passage to the Discussion on Page 16 line 4 detailing that nutrients may be released from shrunken lipid droplets via magnesium treatment and supplied to mitochondria, which may promote hyphal growth in *T. asahii* and *T. coremiiforme*.

3. In figures 4-8, the authors should directly indicate what the colors represent in

the figure itself, rather than only in the figure legend. This will help clarify the interpretation of the data for readers.

- Simple explanations regarding the orange, blue, and gray bars were added in Figures 4, 6, 7, and 8.

4. A helpful tip for future studies would be for the authors to test the incubation time with BODIPY when staining LDs to optimize the staining conditions for the best results.

- We carried out BODIPY staining with the incubation times of 0, 1, 2, 5, and 10 min to examine the optimal incubation time using *A. gamsii* as a representative yeast. However, BODIPY signals were bright enough even after 1 min incubation and the brightness of BODIPY signals did not change with increasing incubation time from 1 to 10 min. The result was added on Page 20 line 13 in the Cell staining section of MATERIALS AND METHODS and is also described in Fig. S5. Therefore, we think that lipid droplets were sufficiently stained by 5 min incubation with BODIPY in the study.

To Reviewer 2

Reviewer 2 (Comments for the Author):

I would prefer if the author can add a special section for statistical analysis.

- We appreciate the kind suggestion that the Statistical analysis section is missing in the paper. We added the following passage on Page 22 line 9 in the MATERIALS AND METHODS: ‘Statistical differences between samples were calculated using the Mann-Whitney U Test with two-tailed hypothesis. Each data set was introduced into the calculator on the website (<https://www.socscistatistics.com/tests/mannwhitney/default2.aspx>) and the statistical significance for each was calculated. The p -values are significant at $p < 0.05$.’

Re: Spectrum03210-24R1 (Analyses of hyphal diversity in Trichosporonales yeasts based on fluorescent microscopic observations)

Dear Dr. Keita Aoki:

Congratulations!

Your manuscript has been accepted, and I am forwarding it to the ASM production staff for publication. Your paper will first be checked to make sure all elements meet the technical requirements. ASM staff will contact you if anything needs to be revised before copyediting and production can begin. Otherwise, you will be notified when your proofs are ready to be viewed.

Sincerely,
Michael Klutstein
Editor
Microbiology Spectrum

Reviewer #1 (Comments for the Author):

I appreciate the authors' response and have no additional questions.